# Involvement of Cutaneous Sensory Corpuscles in Non-Painful and Painful Diabetic Neuropathy

**DOI:** 10.3390/jcm10194609

**Published:** 2021-10-08

**Authors:** Yolanda García-Mesa, Jorge Feito, Mario González-Gay, Irene Martínez, Jorge García-Piqueras, José Martín-Cruces, Eliseo Viña, Teresa Cobo, Olivia García-Suárez

**Affiliations:** 1Grupo SINPOS, Departamento de Morfología y Biología Celular, Universidad de Oviedo, 33006 Oviedo, Spain; garciamyolanda@uniovi.es (Y.G.-M.); jfeito@saludcastillayleon.es (J.F.); garciapiquerasjorge@gmail.com (J.G.-P.); pepe3214@gmail.com (J.M.-C.); eliseovina@telecable.es (E.V.); 2Servicio de Anatomía Patológica, Complejo Asistencial Universitario de Salamanca, 37007 Salamanca, Spain; 3Sercivio de Angiología y Cirugía Vascular, Fundación Hospital de Jove, 33290 Gijón, Spain; mariogg75@hotmail.com; 4Sercivio de Cirugía Plástica y Reparadora, Fundación Hospital de Jove, 33290 Gijón, Spain; ire.garciamartinez@gmail.com; 5Servicio de Cardiología, Unidad de Hemodinámica y Cardiología Intervencionista, Hospital de Cabueñes, 33206 Gijón, Spain; 6Departamento de Cirugía y Especialidades Médico-Quirúrgicas, Universidad de Oviedo, 33006 Oviedo, Spain; teresacobo@uniovi.es

**Keywords:** distal diabetic sensorimotor polyneuropathy, painful and non-painful distal diabetic sensorimotor polyneuropathy, cutaneous sensory corpuscles, human glabrous skin, mechanoproteins

## Abstract

Distal diabetic sensorimotor polyneuropathy (DDSP) is the most prevalent form of diabetic neuropathy, and some of the patients develop gradual pain. Specialized sensory structures present in the skin encode different modalities of somatosensitivity such as temperature, touch, and pain. The cutaneous sensory structures responsible for the qualities of mechanosensitivity (fine touch, vibration) are collectively known as cutaneous mechanoreceptors (Meissner corpuscles, Pacinian corpuscles, and Merkel cell–axonal complexes), which results are altered during diabetes. Here, we used immunohistochemistry to analyze the density, localization within the dermis, arrangement of corpuscular components (axons and Schwann-like cells), and expression of putative mechanoproteins (PIEZO2, ASIC2, and TRPV4) in cutaneous mechanoreceptors of subjects suffering clinically diagnosed non-painful and painful distal diabetic sensorimotor polyneuropathy. The number of Meissner corpuscles, Pacinian corpuscles, and Merkel cells was found to be severely decreased in the non-painful presentation of the disease, and almost disappeared in the painful presentation. Furthermore, there was a marked reduction in the expression of axonal and Schwann-like cell markers (with are characteristics of corpuscular denervation) as well as of all investigated mechanoproteins in the non-painful distal diabetic sensorimotor polyneuropathy, and these were absent in the painful form. Taken together, these alterations might explain, at least partly, the impairment of mechanosensitivity system associated with distal diabetic sensorimotor polyneuropathy. Furthermore, our results support that an increasing severity of DDSP may increase the risk of developing painful neuropathic symptoms. However, why the absence of cutaneous mechanoreceptors is associated with pain remains to be elucidated.

## 1. Introduction

Diabetic neuropathy includes a group of neuropathies associated with diabetes mellitus, which are the main cause of morbidity and mortality in these patients. The most common complication during the evolution of type 2 diabetes mellitus is distal diabetic sensorimotor polyneuropathy (DDSP) which may affect up to 50% of patients [1], leading to neuropathic pain in as many as 50% of patients [2,3]. The Toronto Consensus (2011) [2] defined DDSP pain as “*pain that is a direct consequence of abnormalities in the peripheral somatosensory nervous system in diabetic individuals*”. Diagnostic of DSSN is frequently delayed due to the scarcity of early diagnostic tests, and recently, axonal swellings in cutaneous biopsies have been proposed as an early marker of sensory nerve injury in type 2 diabetes mellitus [4,5]. Consequently, there is a high risk of ulceration with subsequent distal amputation in lower limbs [6,7]. During the last decade, diagnostic tests such as skin biopsy and corneal confocal microscopy have confirmed its usefulness in the diagnosis of peripheral neuropathies through quantitative analysis of Aδ and C fine intraepithelial nerve fibers [3,8,9,10,11]. Thus, more severe small fiber damage in the skin of patients with painful diabetic neuropathy compared with painless diabetic neuropathy has been observed, and the density of intraepithelial nerve fibers was lower in subjects with painful compared with painless neuropathy [11,12].

In addition to pain, light touch and low-frequency vibration are also impaired in DDSP [13,14,15], suggesting involvement of Aβ fibers and sensory corpuscles (Meissner and Pacinian corpuscles, Merkel cell–neurite complexes; see [16]). Thus, the analysis of sensory corpuscles in cutaneous biopsies has been proposed as a “gold standard” method of diagnostic interest in some peripheral neuropathies and neurodegenerative diseases (see for a review [17,18]).

Focusing on diabetic neuropathy, the seminal article of Ras and Nava [19] in diabetic mice demonstrate a decrease in the number in Meissner-like corpuscles as well as axonal changes. Some years later, Paré et al. [20] observed two phases in corpuscular deterioration in diabetic monkeys. During the first phase, they found a hypertrophy of Meissner corpuscles and Merkel endings, followed by a second phase in which the number of corpuscles declined (but remained higher than in age-matched nondiabetic animals) and the Merkel innervation was reduced (to age-matched nondiabetic levels). Furthermore, the diabetic Meissner corpuscles had an abnormal structure and immunochemistry and Pacinian corpuscles also deteriorated. Data from human skin biopsies revealed a reduction in both Meissner corpuscles and their afferent Aβ myelinated nerve fibers which correlated with decreased amplitudes of sensory/motor responses [21]. Additionally, using in vivo reflectance confocal microscopy, Meissner corpuscles were found reduced in density in diabetes relative to controls [22]. Nevertheless, to the best of our knowledge, a detailed study of human cutaneous sensory corpuscles in painful and non-painful DDSN has not been performed.

Thus, the present study was designed to analyze cutaneous sensory corpuscles from the feet of subjects undergoing painful and non-painful DDSN. We investigated changes in the density, size axonal, and Schwann-like cells of Meissner and Pacinian corpuscles as well as Merkel cell–neurite complexes. Furthermore, we analyzed possible changes in putative mechanoproteins detected on sensory corpuscles [23] which could be at the basis of mechanosensory impairment in DDSN.

## 2. Materials and Methods

### 2.1. Patients

Subjects of both genders, free of neurologic disease, who suffered accidental toe amputation (*n* = 10) were used as the control group. Patients clinically and analytically diagnosed with DM with non-painful (*n* = 10) or with painful diabetic neuropathy (*n* = 10) who were subjected to toe amputation due to ischemic complications of DM were also studied. The control skin samples were collected within 6 h after incidental toe amputation at the Service of Plastic Surgery of the Hospital Universitario Central de Asturias, Oviedo, Principality of Asturias, Spain. The skin samples from diabetic patients were collected within 3 h after amputation and were obtained at the Service of Vascular Surgery, Fundación Hopital Jove of Gijón, Principality of Asturias, Spain. The age range of the subjects was 48 to 84 years. The study was approved by the Ethical Committee for Biomedical Research of the Principality of Asturias, Spain (Cod. CElm, PAst: Proyecto 266/18). All materials were obtained in compliance with Spanish law (RD 1301/2006; Ley 14/2007; DR 1716/2011; Orden ECC 1414/2013), and according to the guidelines of the Helsinki Declaration II.

Data managing of the diabetic subjects included in the study were divided into 5 parts as follows:Clinical history: age, gender, time of evolution of disease, variant of peripheral neuropathy, HbA1c value, presence of proinflammatory and inflammatory factors (c-reactive protein, erythrocyte sedimentation rate), alterations in blood clotting test, ankle-brachial index, Doppler, and nervous conduction studies.Physical examination: maintained local sensibility, popliteal artery pulse assessment, skin alterations or deformities and allodynia/hyperalgesia/paresthesia/anesthesia. Sensitivity was focused on the clinical examination if anatomical structures and biochemical channels in study are responsible of this sensation.Monofilament testing: to ascertain the presence of sensibility in 4 random points at the affected extremity.Plantar discrimination: capacity, which is closely associated with mechanosensory receptors. Alterations in this variable may be related to an increase, decrease, or absence of mechanoreceptor.DN4 test to estimate neuropathic pain: Previous studies describe how alterations on these sensory structures may produce extreme effects in the form of a total anesthesia in the studied region, or even an excessive painful response under normally painless stimuli. Data from patients and analytical are summarized in Table 1.

The patients included in the study were under different treatments (active principles are in brackets). Those suffering from non-painful DDSP received Bemiparin, Metformin, Efficib (metformin + sitagliptin), Repaglinid, Neparvis (Valsartan + sacubitril), Glargina Insulin sc; those undergoing painfull DDSP received Velmetia (metformin + sitagliptin), Humalog 200 UI/mL (lispro insulin), Toujeo 300 UI/mL (glargine insulin), Diamicron (gliclazide), Dianben (metformin), Trulicity (dulaglutide). Additonal treatment in non-painful DDSP were anticoagulants and antiagregants: Adiro (acetilsalicilic acide), Sugiran (prostaglandin E1), Trinomía (atorvastatin, acetilsalicilic acide and ramipril), Emcorcor (bisoprolol), Digoxina, Atorvastatin; antihypertensive: Irbersartan, Estatin, Furosemida, Enalapril); and anxiolytics: Lorazepam. Additional treatment in P DDSP were anticoagulants and antiagregants: Pradaxa (dabigatran etexilate), Adiro (acetilsalicilic acide), Atorvastatin, antihy-pertensive (Atenolol, Enalapril, Bisoprolol (bisoprolol fumarato); anxiolytic: Ansium (sulpiride + diazepam), Lorazepan; when analgesic drugs were required also, Paracetamol and Metamizol (magnesium metamizole) were administered.

### 2.2. Material and Treatment of the Tissues

Skin samples (*n* = 30), 0.5 × 1 × 0.c cm approximately perpendicular to the skin surface were obtained from the plantar aspect of the distal phalanx of toes. The specimens were fixed in 4% formaldehyde in 0.1 M phosphate-buffered saline (pH 7.4) for 24 h, dehydrated and routinely processed for paraffin embedding.

### 2.3. Histology and Immunohistochemistry

*Hematoxylin-Eosin*—Deparaffinized and rehydrated sections were introduced 10 min into Harris Hematoxylin. Subsequently, they were washed in water and passed for 5 s by acidic water (with glacial acetic acid), and finally in an eosin solution for 30 s. The sections were washed, dehydrated, diaphanized in Xylol and mounted with Entellan^®^.

*Single immunohistochemistry*—Deparaffinized and rehydrated sections were processed for indirect detection of antibodies (see Table 2), using the EnVision antibody complex detection kit (Dako, Copenhagen, Denmark), following supplier’s instructions. Briefly, the endogenous peroxidase activity was inhibited (3% H_2_O_2_ for 15 min) and the non-specific binding was blocked (10% bovine serum albumin for 20 min). Sections were then incubated overnight at 4 °C with the primary antibody. Subsequently, sections were incubated with the anti-rabbit and anti-mouse EnVision system-labelled polymer (Dako-Cytomation, Santa Clara, USA) for 30 min, washed in buffer solution, and treated with peroxidase blocking buffer (Dako Cytomation). Finally, the slides were washed with buffer solution and the immunoreaction was visualized with diaminobenzidine as a chromogen, then washed, dehydrated, and mounted with Entellan^®^ (Merk, Darmstadt, Germany). To ascertain structural details, the sections were counterstained with Mayer’s hematoxylin.

*Double immunofluorescence*—Sections were also processed for simultaneous detection of PIEZO2, ASIC2 and TRPV4 together with specific markers for Schwann-like cells (S100 protein), axons (Cobo et al., 2021), and for Merkel cells [24,25] (see Table 1). Non-specific binding was reduced by incubating the sections for 30 min with a solution of 25% calf bovine serum in tris buffer solution (TBS). The sections were incubated overnight at 4 °C in a humid chamber with a 1:1 *v*/*v* mixture of the polyclonal antibody against PIEZO2, ASIC2 and TRPV4 with monoclonal antibodies against S100 protein, neurofilament protein (NFP), neuron-specific enolase (NSE), cytokeratin 20 (CK20) and chromogranin A (ChrA). After rinsing with TBS, the sections were incubated for 1 h with CFL488-conjugated bovine anti-rabbit IgG (sc-362260, Santa Cruz Biotechnology, Heidelberg, Germany), diluted 1:200 in TBS, them rinsed again and incubated for another hour with CyTM3-conjugated donkey anti-mouse antibody (Jackson-ImmunoResearch, Baltimore, MD, USA) diluted 1:100 in TBS. Both steps were performed at room temperature in a dark humid chamber. Sections were finally washed, and the cell nuclei were stained with DAPI (10 ng/mL). Triple fluorescence was detected using a Leica DMR-XA automatic fluorescence microscope (Microscopía fotónica y Proceso de imágen, Servicios científico-técnicos, Universidad de Oviedo) coupled with a Leica Confocal Software, version 2.5 (Leica Microsystems, Heidelberg GmbH, Germany) and the images captured were processed using the software Image J version 1.43 g Master Biophotonics Facility, Mac Master University Ontario (www.macbiophotonics.ca (access on 11 January 2021)).

For control purposes, representative sections were processed in the same way as described, using non-immune rabbit or mouse sera instead of the primary antibodies or omitting the primary antibodies in the incubation. Furthermore, when available, additional controls were carried out using specifically preabsorbed antisera. Under these conditions, no positive immunostaining was observed (data not shown).

### 2.4. Quantitative Study

Quantitative analyses were performed to determine the density of cutaneous digital sensory corpuscles. We examined 500 sections of glabrous toe skin from controls (*n* = 100), NP DDSP (*n* = 200) and P DDSP (*n* = 200) subjects, to evaluate the density of sensory corpuscles. The number of Merkel cells, as well as Meissner and Pacini corpuscles was calculated as follows: 20 fields were quantified in microscopy at 10×, per individual, in 5 sections separated by 50 µm, by two different observers, and the results obtained were averaged. Data are expressed as mean ± SD/mm^2^. In turn, to verify the functionality of the nervous structures under study, the percentage of positive PIEZO2 mechanoreceptors was estimated, performing the quantification with double immunofluorescence as described below: the percentage of presumably functional Merkel cells was quantified by the simultaneous detection of PIEZO2 and CK20 (as specific marker) in 5 sections separated from each other by 50 µm, while for the percentage of positive Meissner and Pacinian PIEZO2 corpuscles, the PIEZO2 and S100P antibodies (as specific marker for Schwann cells) were used, using the following formula: % PIEZO2+: (Average number of total PIEZO2 + mechanoreceptors/Average number of total mechanoreceptors) × 100.

## 3. Results

### 3.1. Quantitative Analyses of Cutaneous Sensory Corpuscles: Association between Density and Neuropathy

A total of 168 Meissner corpuscles were analyzed, all of them identified since the expression of S100P by the Schwann-like (lamellar) cells. Meissner corpuscle density was lower in neuropathy patients compared with controls: there was a reduction of about 80% in NP DDSP, and an almost complete absence in P DDSP (only 2 Meissner corpuscles were identified in this group of patients) (Figure 1).

The results obtained in Pacinian corpuscles (identified because of morphology and the occurrence of S100P in the Schwann-like cells; *n* = 58) were parallel to those of Meissner corpuscles: there was a reduction of about 50% in NP DDSP, and an almost complete absence in P DDSP (Figure 1).

Regarding Merkel cells (identified by the expression of CK20), significant differences were also found between the controls and DDSP. In the NP DDSP, there was a reduction of about 68% with respect to the controls, and in P DDSP the reduction reached 93%. However, 7 patients from this group maintained a density of Merkel cells similar to the controls (Figure 1).

### 3.2. Immunohistochemical Profile of Meissner and Pacinian Corpuscles

Meissner corpuscles were studied for detection of the axonal markers NFP and NSE, and Schwann-like cells for detection of S100P (see [16]). Meissner corpuscles were identified at all three groups analyzed, although in P DDSP they were found in only 2/10 subjects.

Differences in the morphology, size, number, placement within dermis, and intensity of immunostaining were observed among the three groups evaluated. In the control group, Meissner corpuscles were elongated and were always localized in the dermal papillae. The lamellar cells were packed, arranged in parallel, and displayed strong S100 protein immunoreactivity (Figure 2a,b). The axons showed immunoreactivity for NFP which run all over the corpuscle (Figure 2c), in an irregular course among lamellar cells (Figure 2d). In the NP DDSP patients, the predominant localization of Meissner corpuscles also was within the dermal papillae, but some were also displaced to the dermis (Figure 2e). Furthermore, the size was reduced, and the morphology was rounded. The lamellar cells were disorganized and unstructured form (Figure 2e,f), and the axon was sometimes unidentified (Figure 2g,h). The scenario changed dramatically in the P DDSP group in which Meissner corpuscles were not identified, and when identified morphologically they lacked S100P and NF protein (Figure 2i–k). Interestingly, in the dermis of P DDSP, abundant S100P-positive dendritic cells were observed (Figure 2i,j).

Mechanotransduction is the process which converts mechanical forces in action potentials, and in the skin, it occurs in mechanoreceptors, including Meissner and Pacinian corpuscles, and Merkel cell–neurite complexes (see [23]). In this process, some mechano-gated ion channels such as ASIC2, TRPV4, and, primarily, PIEZO2 [23] participate. Thus, we have investigated the possible changes in these mechanoproteins within sensory corpuscles during DDSP as a possible partial explanation of touch changes found in this disease. Using double immunofluorescence associated with confocal microscopy, we observed the occurrence of PIEZO2 (Figure 3a), ASIC2 (Figure 3d), and TRPV4 (Figure 3g) restricted to the axon, but not in the lamellar cells of Meissner corpuscles of control subjects. Immunofluorescence for all three mechanoproteins investigated was absent from Meissner corpuscles of both NP DDSP (Figure 3b,e,h) and P DDSP (Figure 3c,f,i).

Evident changes were also noted in Pacinian corpuscles of DDSP patients with respect to the controls (Figure 4). The Pacinian corpuscles from normal subjects showed the typical onion-layer arrangement. The axon was placed at the center (displaying NFP immunoreactivity) of the corpuscle and was encircled by the lamellae of the Schwann-like cells that form the inner core (positive for S100 protein); outside these neural components are the intermediate layer, the outer core, and the capsule derived from the endoneurium and the perineurium (Figure 4a–c). In subjects with NP DDSP, the neural compartment of the Pacinian corpuscles (i.e., the axon and the inner core) was disarranged (Figure 4d,e), and no immunoreactivity for NFP was detected, nor in most axons (it was present in about 10%) or S100 protein (it was detected in about 15%). Nevertheless, the Pacinian corpuscles can be identified based on their morphology. Finally, in the Pacinian corpuscles from P DDSP patients, no immunoreactivity was detected for NFP or S100 protein (Figure 4g–i).

Regarding the investigated mechanoproteins PIEZO2, ASIC2, and TRPV4 they were detected restricted to the axon of Pacinian corpuscles from control subjects. Conversely, they were absent from the axon of both NP DDSP and P DDSP (Figure 5).

### 3.3. Immunohistochemical Profile of Merkel Cells

Merkel cells are epidermal cells localized in the basal stratum of the epidermis that is selectively immunolabelled with antibodies against cytokeratin 20 (CK20) or chromogranin A (ChrA). Using the expression of these proteins as a marker, Merkel cells were found in the epidermal rete pegs of all subjects investigated but the number was reduced in DDSP patients, and the number of cells decrease with neuropathic (Figure 6 and Figure 7). In the skin of control subject clusters of up to four Merkel cells were regularly observed (Figure 6a,d,g,j), whereas in NP DDSP (Figure 6b,e,h,k) and P DDSP patients (Figure 6c,f,i,l) Merkel cells were found scattered in the basal stratum of nerve-formed clusters. Nevertheless, the remaining cells retained the basic immunohistochemical profile for CK20 or ChrA.

To investigate whether Merkel cells were mechanosensitive, we analyzed the occurrence of PIEZO2 within them [26]. PIEZO2 was observed co-localized with CK20 (Figure 6g–i)- and ChrA (Figure 6j–l)-positive cells, thus were identified as Merkel cells. However, the density of Merkel cells and PIEZO2-positive Merkel cells progressively decreased with the evolution of disease (Figure 7).

## 4. Discussion

The present study was designed to investigate the changes in cutaneous sensory corpuscles and Merkel cell during diabetic neuropathic. This topic has been partially analyzed although it is broadly accepted that diabetic neuropathy is accompanied by a progressive impairment in the somatosensory system that affects the quality of life of patients [27,28]. This deterioration involves all levels of the somatosensory pathways from the skin with the cutaneous peripheral somatosensory receptors (free nerve endings, sensory corpuscles, Merkel cell–axonal complexes); see [16,29] for the cerebral cortex [28,30,31]. In the study we used skin to samples from control subjects as well as from patients suffering NP DDSP and P DDSP to analyze the cutaneous Meissner and Pacinian corpuscles, as well as Merkel cells–axonal complexes. We used immunohistochemistry associated with a battery of antibodies to identify the axon and Schwann related cells (since both are involved in diabetic neuropathy [32,33]) and a series of putative mechanoproteins (PIEZO2, ASIC2 and TRPV4) which are involved in mechanotransduction [23].

The effects of diabetic neuropathy on sensory loss and pain have been reviewed in detail by Shillo et al. [30] and Rosenberger et al. [31], and although most studies support a correlation between neuropathy severity and neuropathic pain in DDSP [34,35], others do not [36]. In any case, the evidence suggests that an increasing severity of DDSP may increase the risk of developing painful neuropathic symptoms thus frequently evolving to P DDSP [2,37].

It is now accepted that cutaneous sensory corpuscles share the cell composition and immunohistochemical profile of the sensory fibers of which they depend. Therefore, their evaluation through cutaneous biopsy can be a useful method to evaluate and follow-up DDSP evolution and/or treatment. Early studies have reported a decrease in the density and structural deterioration of sensory corpuscles in diabetic monkeys [20] (Paré et al., 2007) and humans [21]. As far as we know, a detailed study in humans differentiating between NP DDSP and P DDSP was never carried out.

On the other hand, studies on human sensory corpuscles in DDSP are limited and do not use specific immunohistochemistry assays for the neural component (i.e., axon and Schwann-related cells). In the present study, we observed that progression from NP DDSP to P DDSP courses with a reduction in size of Meissner corpuscles, changes in the morphology and cellular arrangement, displacement to deep dermis, and loss of immunoreactivity for axonal and Schwann-like cell markers (Figure 8). The reduction or absence of S100 protein in lamellar cells, together with absence of NFP immunostaining, strongly suggest denervation of those corpuscles [38,39,40].

Our results on NP DDSP are in good agreement with those reported by Peltier et al. [31] in the digital glabrous skin. These authors observed that in diabetes type II, Meissner corpuscles decrease and become disorganized, and these changes are attributed to axonal loss. However, to our knowledge, the absence of Meissner corpuscles that occurs in P DDSP has not been previously reported. Thus, it seems that the transition from non-painful to painful DDSP is due to a loss of Meissner corpuscles and therefore a loss in tactile discrimination.

Regarding Pacinian corpuscles, the effects of DDSP were evident in both number and structure and are consistent with the data reported by Pare et al. [30] in Pacinian corpuscles of diabetic monkey. These authors observed that Pacinian corpuscles showed a pronounced disruption consisting of breakdown in the inner lamella, irregular spacing between lamellae, and thickening of the outer lamellae. In the present study, we have found a marked decrease in the number of Pacinian corpuscles associated with diabetes neuropathic, in fact a dramatic reduction was observed in P DDSP. It must be noted that Pacinian corpuscles of NP DDSP patients showed inner core destruction whereas painful-DPN patients both the inner core and the complex outer core/capsule are affected by disease (Figure 8). As far as we know, these changes have not been reported earlier.

In relation with Merkel cells our results disagree with those from Pare et al. [30]. These authors reported an increase in Merckel cells in the shorter-term hyperglycemic monkeys, whereas we have found an important reduction in number of Merkel’s cells associated with DDSP, especially P DDSP. Furthermore, this condition results in absence of the typical clusters of Merkel cells in normal skin.

During the last decade it has been definitively established that the different qualities of somatosensitivity depend on the expression of different ion channels in specific subtypes of primary sensory neurons and their peripheral terminals. In particular, PIEZO2, ASIC2, TRPV4, and TRPC6, participate in different modalities of mechanosensitivity (see for a review [23]). Previous studies from our laboratory have demonstrate the occurrence of PIEZO2, ASIC2, TRPC6 and TRPV4 in the axons, of ASIC2 and TRPV4 in the corpuscular Schwann-related cells, and of PIEZO2 and ASIC2 in the Merkel cell–neurite complexes. Our results in the skin of control subjects are in complete agreement with previous data. Interestingly, all those proteins disappeared from the axon of both groups of DDSP, thus lending further support to the observation of absence of NFP-positive axons, and to the tactile impairment of touch in diabetic patients (Figure 8). Nevertheless, further studies are necessary to confirm the correlation between mechanoproteins depletion in cutaneous mechanoreceptors and tactile and vibration alterations in diabetes [13,30,31].

## 5. Conclusions

Overall, present results demonstrate that Meissner and Pacinian corpuscles and Merkel cells (a part of the Merkel cell–axonal complex), which represent the most peripheral part of the mechanosensory system, undergo progressive topographical, morphological, and structural changes from non-painful to painful DDSP. These changes also affect the expression of axonal and Schwann-like cell proteins, as well as some putative mechanoproteins. These variations are in good agreement with previous experimental studied but in those studies a differential between non-painful and painful DDSP cannot be established. Furthermore, corpuscular alterations observed here can explain the tactile defects associated with DDSP, but not the presence of pain. Nevertheless, our results support that an increasing severity of DDSP may increase the risk of developing painful neuropathic symptoms. Nociception is associated with free nerve endings [29], which were not investigated in this study and have been found reduced in diabetic neuropathy [11,12]. In our opinion the occurrence of pain when mechanoreceptors deteriorate could be related with changes in the circuitry of the dorsal horn of the spine affecting the gate-control of pain (predominance of nociceptive inputs vs. decrease in mechanical inputs) or changes in non-neuronal cells especially microglia [41,42]. Nevertheless, a contribution of local ischemia to deterioration of sensory corpuscles and pain cannot be ruled out.

## Figures and Tables

**Figure 1 jcm-10-04609-f001:**
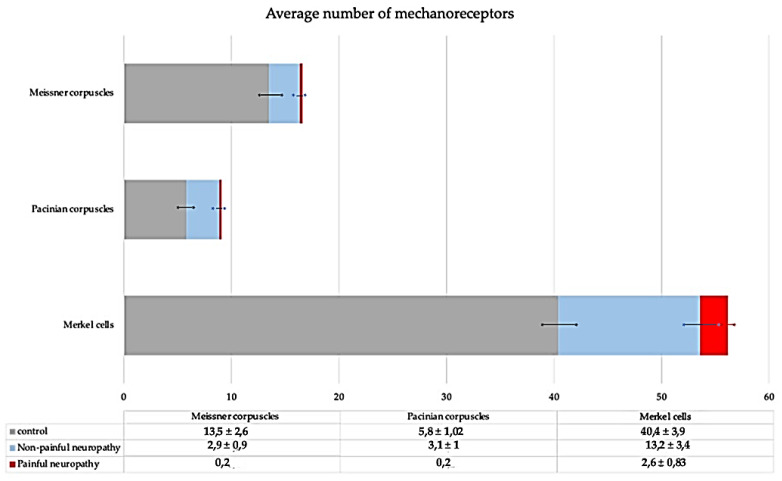
Average number of Meissner corpuscles, Pacinian corpuscles, and Merkel cells in the glabrous skin of the toes in control (grey), non-painful DPN (blue), and painful DPN (red) for 20 fields at 10×. Data are expressed as mean ± SD/mm^2^. A reduction in the number of all types of sensory corpuscles studied was detected with disease, in fact, the most dramatic decrease was in painful-DPN subjects. * is S.D.

**Figure 2 jcm-10-04609-f002:**
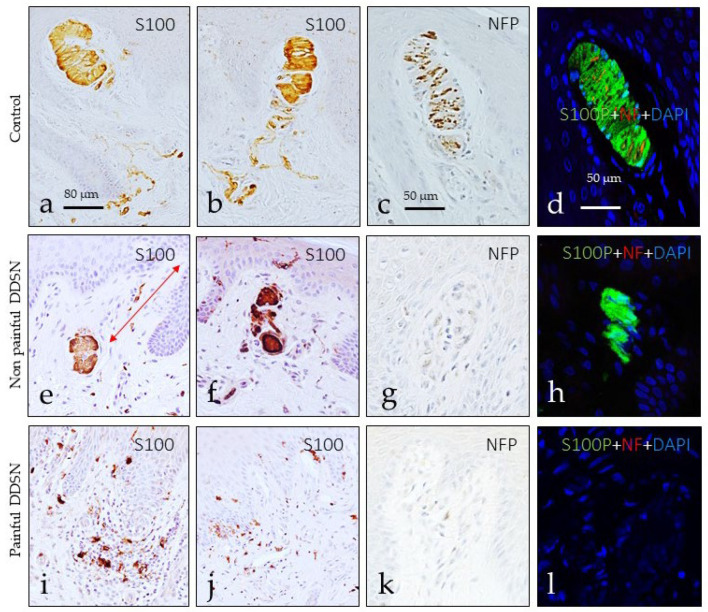
Meissner corpuscles from the glabrous toe skin of subjects belonging to each group studied. S100 protein (S100P) was used to immunolabel the lamellar cells, and neurofilament protein to immunolabel the axons. In controls, Meissner corpuscles were located inside the dermal papillae, and both the lamellar cells and the axons displayed a typical aspect (**a**–**d**). The Meissner corpuscles from NP DDSP subjects were mainly placed in the dermis, outside the dermal papillae, were smaller and the lamellar cells disarranged while no-positivity for NFP was detected (**e**–**h**). In P DDSP Meissner corpuscles were lost (**i**–**l**) and large infiltrated of dendritic S100P-positive cells were observed. S100: S100 protein; NFP: neurofilament protein; NF: neurofilament; DAPI: 4′,6-diamidino-2-phenylindole.

**Figure 3 jcm-10-04609-f003:**
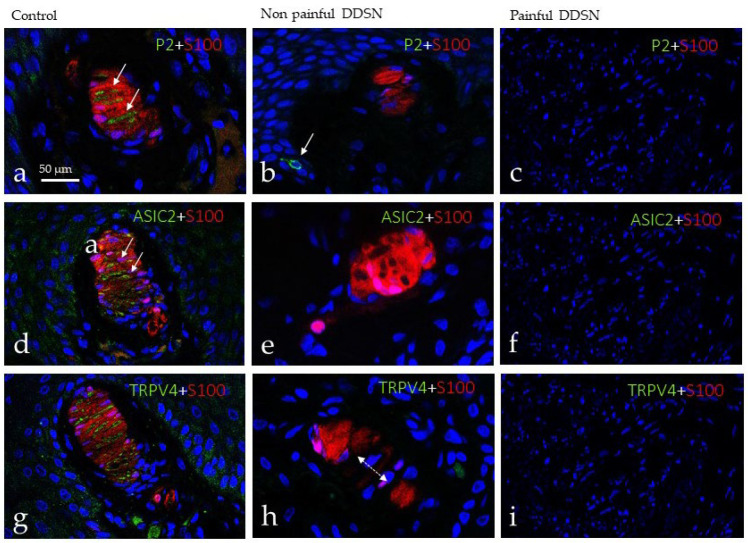
Double immunofluorescence for S100 protein (red fluorescence) and PIEZO2, ASIC2 and TRPV4 (green fluorescence) in Meissner corpuscles of glabrous toe skin of control (**a**,**d**,**g**), NP DDSP (**b**,**e**,**h**), and P DDSP (**c**,**f**,**i**) patients. Sections were counterstained with DAPI to ascertain structural details. In controls PIEZO2, ASIC2 and TRPV4 were regularly detected in the axons, were all these mechanoproteins were undetectable in the corpuscles of NP DDSP and P DDSP patients. P2: Piezo 2; ASIC2: Acid sensing-ion channel; TRPV4: Trasient receptor potential cation channel V4.

**Figure 4 jcm-10-04609-f004:**
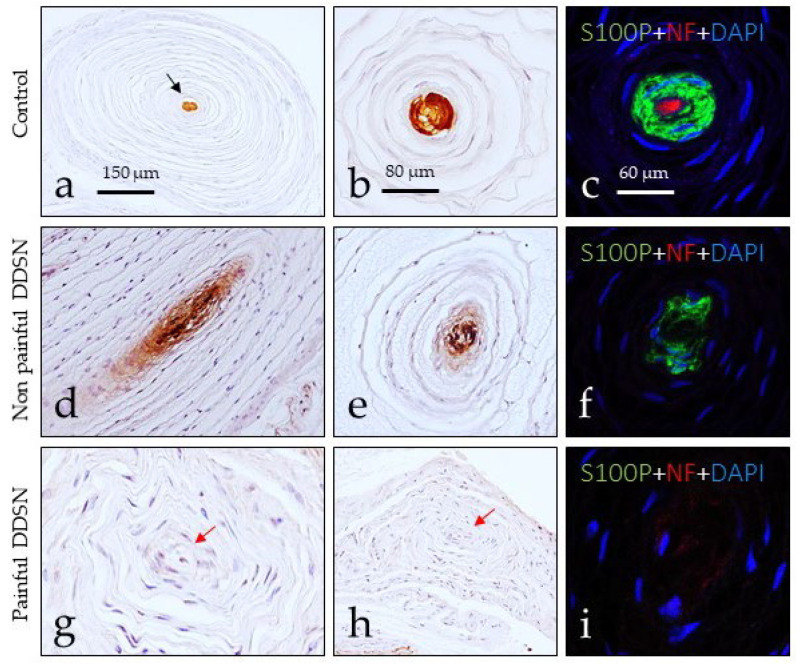
Immunohistochemical localization of S100 protein (S100P) in the inner core cells and neurofilament protein (NFP) in axon of human toes Pacinian corpuscles from control (**a**–**c**), NP DDSP (**d**–**f**) and P DDSP (**g**–**i**) patients. The axon was identifiable only in control group (red fluorescence **c**,**f**,**i**) while S100 protein (S100P) immunolabeling was found in the inner core of control and NP DDSP patients whereas it was undetectable in P DDSP (red arrows in (**g**,**h**)).

**Figure 5 jcm-10-04609-f005:**
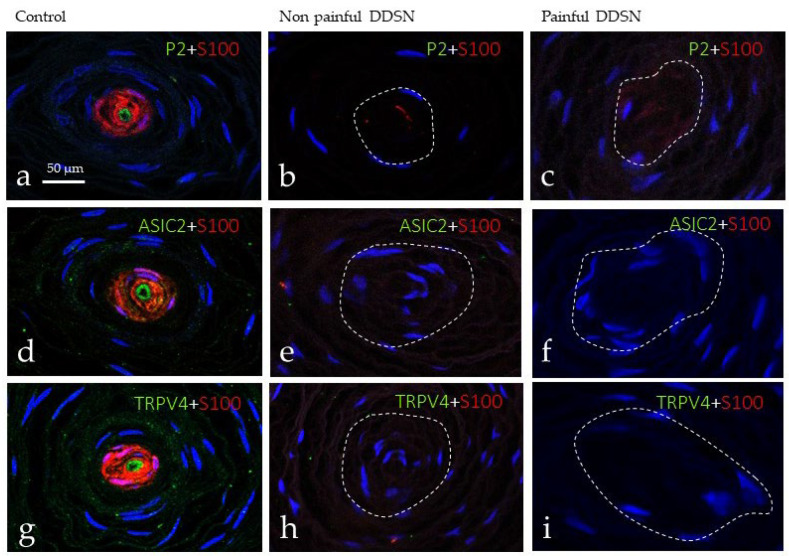
Double immunofluorescence for S100 protein, PIEZO2 (P2; (**a**–**c**)), ASIC2 (**d**–**f**) and TRPV4 (**g**–**i**) in cutaneous Pacinian corpuscles from control (**a**,**d**,**g**), NP DDSP (**b**,**e**,**h**), and P DDSP (**c**,**f**,**i**) patients. PIEZO2, ASIC2, and TRPV 4 were presented only in the axon of Pacinian corpuscles.

**Figure 6 jcm-10-04609-f006:**
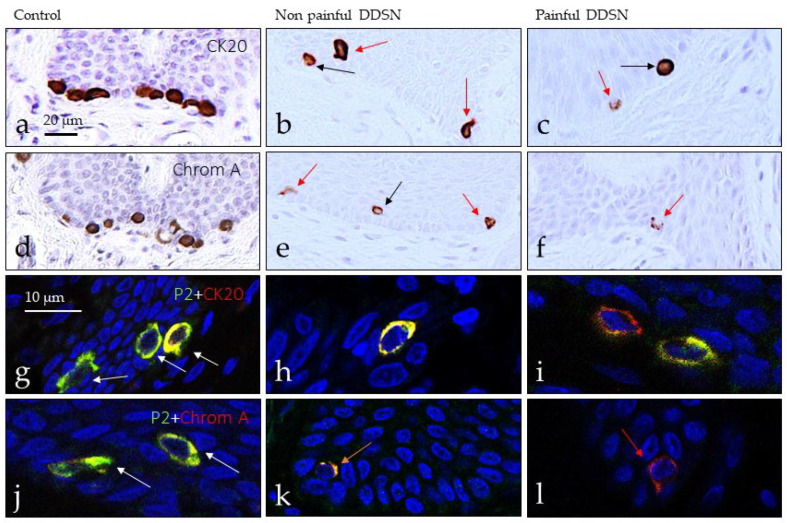
Cutaneous Merkel cells were identified since the localization within the basal stratum of the epidermis, and the expression of cytokeratin (CK20) and chromogranin A (ChrA). They were identified in control subjects (**a**,**d**,**g**,**j**) and DDSP patients (**b**,**e**,**h**,**k**); (**c**,**f**,**i**,**l**). In control, Merkel cells formed clusters whereas in diabetic patients were scattered and reduced in number. Merkel cells displayed PIEZO2 in both normal and pathological conditions (**g**–**l**). Ck20: Cytokeratin 20; Chrom A: Chromogranin A.

**Figure 7 jcm-10-04609-f007:**
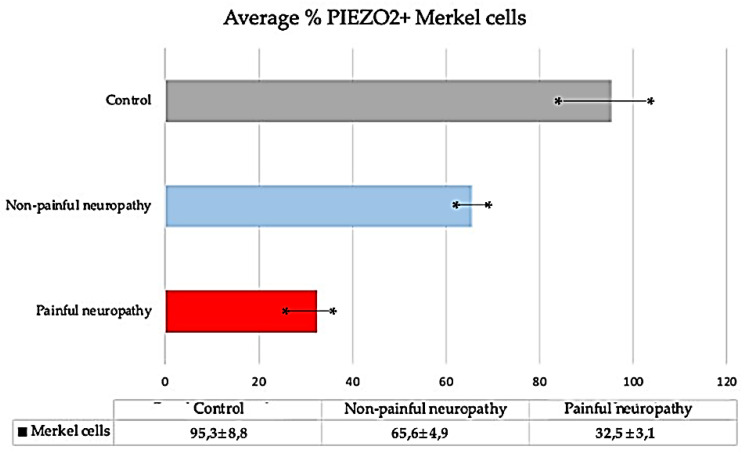
Average (expressed as mean ± DS) of PIEZO2-positive Merkel cells in the skin of control, NP DDSP and P DDSP subjects. A significant reduction in the number of Merkel’s cells was observed with disease. * is S.D.

**Figure 8 jcm-10-04609-f008:**
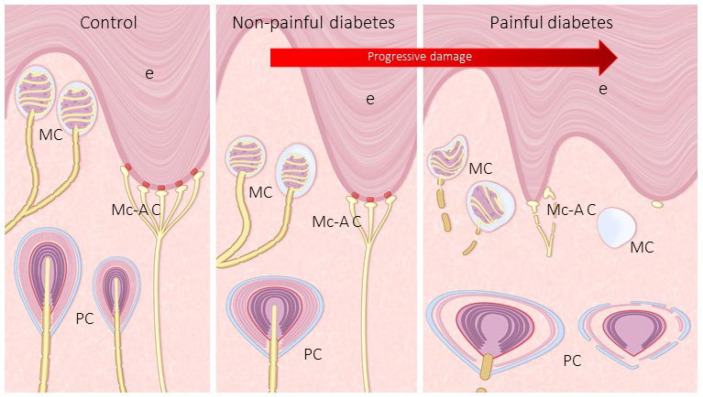
Schematic representation of cutaneous sensory corpuscles (MC: Meissner corpuscles; PC: Pacinian corpuscles) and Merkel cell–axonal complexes (Mc-Ac) in the skin of human toes from control NP DDSP and P DDSP subjects. DDSP results in a reduction in the density of all three types of mechanoreceptors, changes in their placement and morphology, as well as in their immunohistochemical profile and expression of putative mechanoproteins. E:epidermis; PC: pacinian corpuscles.

**Table 1 jcm-10-04609-t001:** Date from patients and analytical.

	Control	NP DDSP	P DDSP
**Age**	62 ± 8.2 s.d.	60 ± 10.3	70 ± 12.4
**Gender**	mixed	mixed	mixed
**Evolution (years)**		11 ± 9.4	21 ± 6.5
**HBA1C**	n.r. (<5.7%)	6.5 ± 0.5	7.6 ± 5.17
**C-reactive protein**	n.r. (<10 mg/L)	16 ± 18.75	39.5 ± 25.14
**GSS**	n.r. (0–29 mm/h)	40.5 ± 58.68	129 ± 76.37
**Prothrombin rate (%)**	n.r. (70–100%)	74.8 ± 40.38	88.6 ± 50.46
**Fibrinogen**	n.r. (200–400 mg/dL)	526.5 ± 376.7	935 ± 198.15
**Ankle-brachial index**	n.r. (0.9–1.3 mmHg)	0.79 ± 0.1	not valuable
**Echo-Doppler**	n.r. (permeable)	permeable popliteal art.	distal obstruction
**Sensitivity**	n.r. (100%)	Yes (100%)	Yes (50%)
**Discriminative capacity**	n.r. (<1 cm)	5.5 cm ± 2.17	8 cm ± 3.09
**Foot pulses**	n.r. (100%)	Yes (33%)	Yes (25%)
**Skin deformity**	n.r. (normal skin)	Yes (50%)	Yes (25%)
**Pain test**	not pain	not pain	electric shock
**Monofilament test**	n.r. (100%)	Positive (50%)	Positive (100%)

NP DDSP: not painful distal diabetic sensorimotor polyneuropathy; P DDSP: painful distal diabetic sensorimotor polyneuropathy; GSS: glomerular sedimentation speed; HBA1C: glycosylated hemoglobin A1C; n.r.: normal range.

**Table 2 jcm-10-04609-t002:** Primary antibodies used in the study.

Antigen (Clone)	Origin	Dilution	Supplier
** *Axonal markers* **			
NSE (BBS/NC/IV-H14)	Mouse	1:100	Dako, Glostrup, Denmark
NFP (NF-H-RNF402)	Mouse	1:200	Santa Cruz Biotechnology, CA, USA
** *Schwann-related cells* **			
S100P	Rabbit	1:5000	Dako, Glostrup, Denmark
S100P (4C4.9)	Mouse	1:1000	Thermo Scientific, Freemont, CA, USA
** *Merkel cells* **			
ChrA (DAK-A3)	Mouse	Prediluted	Dako, Glostrup, Denmark
CK20 (ks 20.8-IS777)	Mouse	Prediluted	Dako, Glostrup, Denmark
** *Ion channels* **			
ASIC 2	Rabbit	1:200	Lifespan Biosciences, Seatle, WA, USA
TRPV4	Rabbit	1:200	Abcamn, Cambridge, UK
PIEZO2	Rabbit	1:500	Sigma-Aldrich, Madrid, Spain

ASIC2: acid-sensing ion channel 2; ChrA: chromogranin A; CK20: cytokeratin20; NFP: neurofilament proteins; NSE: neuron-specific enolase; S100P: S100β protein; TRPV4: transient receptor potential vanilloid 4.

## Data Availability

The data that support the findings of this study are available from the corresponding author upon reasonable request.

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
