# Peer review of "Involvement of Cutaneous Sensory Corpuscles in Non-Painful and Painful Diabetic Neuropathy"

_jcm, 2021, doi:10.3390/jcm10194609_

Round 1

Reviewer 1 Report

This is a well-written article with good overall clinical and scientific importance. The prevalence of polyneuropathy is increasing due to an aging population and the increasing prevalence of risk factors like diabetes mellitus and obesity. Pain in diabetic polyneuropathy is its most costly complication. Also, quality of life is inversely associated with neuropathic pain severity and pain duration in patients with painful polyneuropathy. Therefore the significance of content of this paper is important. We should try to better understand the mechanisms behind painful diabethic neuropathy, to be able to treat the condition better.

I think it is an interesting study and a quite good manuscript with nice pictures. However, I think the manuscript should be revised regarding some aspects I describe below.

ABSTRACT
Quite adequate. I suggest to re-check the conclusions in the abstract. For example, I would recommend adding the sentence in the conclusion of the manuscript text: ‘our results support that an increasing severity of DDSP may increase the risk of developing painful neuropathic symptoms’.

INTRODUCTION
The introduction is adequate with sufficient background and includes relevant references. It has clear structure and a clear objective for the study.

Minor comments:

  • Page 2, line 58: ‘although’: should that not be ‘and’? Does not seem to be a contradisction. Page 2, line 63-64: Thus, the analysis of those formations… What do you mean with formations? For clarity, I suggest to write out what you mean (skin biopsy, enmg?)

METHODS

Comments/suggestions:

  • Page 3: Before the table nothing is described bout the design of the study: especially the control group is not mentioned. How many patients were included? Inclusion criteria? Was informed consent of the patients waived by the Ethical Committee?
  • Page 3, line 124: free of neurologic-disease subjects: is this the control group? I suggest to specify.
  • Page 3, line 130: ‘finger’. Can finger biopsies serve as a control for toe biopsies?
  • Page 3, line 127-128: ‘The skin samples from diabetic patients were collected within 3 h after amputation…’ Do you mean that amputation was an inclusion criteria?
  • Page 3, line 108-109 and 109-110: ‘undergoing’: should be ‘suffering from’ I suppose.
  • Page 4, line 166: NFP, NSE: abbreviations not described in text of manuscript, only under table 2.

RESULTS

Comments/suggestions:

  • Page 5, line 201: Comments: 500 sections from how many patients? From 30 patients?
  • Fig 1: not unit described in the figure?
  • In the results section are some paragraphs about the description of the techniques (e.g. page 8, line 254-260). I am not sure, but they might fit better into the methods section.

DISCUSSION

Comments/suggestions:

  • Page 11, line 340-341: ‘is frequent subtype of peripheral neuropathy defined as’. This sentence is not clear for me..
  • Page 11, line 346-348: ‘Nevertheless, nerve biopsy has been referred to the cutaneous one in valuation these parameters, although some studies have demonstrated decrease in de density of sensory corpuscles and structural deterioration in monkeys’ Again, this sentence is not clear for me in this context.
  • Page 12, line 402-405: ‘In our opinion the occurrence of pain when mechanoreceptors deteriorate could be related with changes in the circuitry of the dorsal horn of the spine affecting the gate-control of pain or changes in non-neuronal cells especially microglia [41,42].’ This is very interesting but not discussed in the discussion section. I suggest to mention this also in the discussion section and explain a bit more your motivations for this opinion.

Minor comments:

  • Page 10, line 318: although it is broadly
  • Page 10, line 322: brackets are not closed
  • Page 11, line 336: neuropathic, do you mean neuropathy?
  • Page 11, line 348: de densitiy, the density?
  • Page 11, line 349: known, know,
  • Page 11, line 361: would not start a sentence with ‘but’
  • Page 11, line 363: ‘to non-painful’, should be ‘from non-painful’
  • Page 11, line 364: ‘is to a loss’, should be ‘is due to a loss’ I suppose
  • Page 11, line 364: ‘therefore in tactile’, is more clear I think if you write: ‘therefore a loss in tactile’
  • Page 12, line 370: decreasing, should be decrease
  • Page 12, line 382-383: ‘For the different modalities of mechanosensitivity PIEZO2, ASIC2, TRPV4 and TRPC6 (see for a review [23]).’ This is not a sentence.

Author Response

To review

On the first place I thank  for your adequate advice. I include in the text all changes that you indicate. 

I must underlie that we only use toes for our research and these samples were always obtained after amputation (accidental in case of control or surgical in case of diabetic patients)

Best regards, 

Olivia García

Reviewer 2 Report

The review titled ‘Involvement of Cutaneous Sensory Corpuscles in Non-painful and Painful Diabetic Neuropathy’ by García-Mesa et al., describes the morphology & density of cutaneous corpuscles, along with the expression of mechano-proteins in sensory corpuscles in skin samples obtained from patients with painful and non-painful distal diabetic sensorimotor polyneuropathy. The main findings of this paper is the disruption in morphology of the cutaneous mechanoreceptors in patients with DDSP. The findings of this paper confirm observations by other groups. The authors also note a topological difference in Pacinian corpuscle destruction between painful and non-painful DDSP samples. The study is well-designed, and the sample number seem adequate. The number of sections per sample, and the number of fields used for analysis also appear adequate.

The following observations were noted:

  1. Table 1: Values depicted are mean, SD? This needs to be better shown
  2. L 123-125: The statement is confusing, please fix
  3. Quantification is robust, but it is important that averages be displayed along with data variability. This is important, considering some of the claims in the article. (Especially in figure 1 and 7. The authors can consider showing the data as bar graph with SD)
  4. Fig 2: To better display what is panel b,f,j. Maybe put this in the legend
  5. Line 256-57: Needs to be fixed
  6. Figure 4: Could the authors quantify? Authors hint that they may have some idea on the percentage of receptors with abnormal staining (Line 279)
  7. In the discussion section, the authors should explain if the duration of diabetes between painful and non-painful DDSP would play a role in study observations. Did the authors see any statistical difference between the two groups?
  8. Ethical approval statement to be included
  9. There are several typographical and grammatical errors in the document. These need to be addressed

Author Response

To reviewer,

First of all I want to thank you for your advice , in fact we have included all your considerations in the text.

We have added DS to the graphs.

Regarding figure2. The letters b,f and I are included in the legend, so (a-d) means (a,b,c and d)

In relation with quantification of Meissner and Paccini  positive immunostaining we added these date in the text , since inmunstaining  practically disappear in pathologic skin and because of that a table was not included. 

Respect to duration of diabetes. The number of individuos in our study was a few so we could not establish a correlation between pain and diabetes time, we need to increase the sample to be able to reach a definitive data.

Reviewer 3 Report

Thank you for the invitation to review this paper with the title “Involvement of cutaneous sensory corpuscles in non-painful and painful diabetic neuropathy”

The paper presents very interesting and novel data on cutaneous sensory receptors in diabetic neuropathy. The main finding of this paper was that the number of sensory corpuscles was decreased in non-painful and painful neuropathy.

As a general comment this paper could benefit from language check which could make the paper more accessible. For example, term “fingers of the foot” is probably a direct translation from the Spanish word. It would be easier to write “toes”. Evolution of disease- do you mean duration of diabetes? Diabetic neuropathic is written often when it should be diabetic neuropathy etc.

Some specific questions and comments:

Line 75 you write “Interestingly, it has been observed more severe small fiber damage in the skin of patients with painful compared to painless diabetic neuropathy, although the density of intraepithelial nerve fibers was lower in subjects with painful compared to painless” – Is this not expected finding that if there is a damage to fibers the density will be lower?

Line 108: Current medication should be presented with generic names.

Table 1:

You probably mean foot pulses?

In that case my question is if the pain could be due to peripheral ischemia rather than diabetic neuropathy? Is it possible that PAD is the reason why you see reduction in the amount of sensory receptors?

Line 123:

Were all subjects (both with or without diabetes) selected after they had suffered accidental amputation or was it just the study subjects without diabetes?

Figure 8. You should rephrase the figure text.

Line 393: “present results demonstrate that Meissner and Pacinian corpuscles, and Merkel cells (a part of the Merkel cell-axonal complex), which represent the most peripheral part of the mechanosensory system, undergo progressive topographical, morphological, and structural changes from non-painful to painful DDSP”

Is it possible to make this kind of statement without having a prospective study? Are there any animal models available?

Author Response

To reviewer,

First of all I want to thank you for your advice we have included all your considerations in the text. 

Respect to prospective study

Our surgeon carried out the test related to pain and sensitivity just before amputation which allow us to know exactly patient's situation , so we did not included a prospective study because this type of test are no carried out in a normal situation. 
